# LogQuant: Log-Distributed 2-Bit Quantization of KV Cache with Superior Accuracy Preservation

## Abstract

We introduce LogQuant, a groundbreaking 2-bit quantization technique for KV Cache in large language model (LLM) inference, delivering substantial memory savings while preserving superior performance. Previous methods either assume that later tokens are more important or attempt to predict important tokens based on earlier attention patterns. Both approaches, however, can result in performance bottlenecks or frequent mispredictions.

LogQuant takes a different approach. By applying a log-based filtering mechanism, it selectively compresses the KV Cache across the entire context, achieving better performance with the same or even reduced memory footprint compared to existing methods. In benchmark tests, it enhances throughput by 25% and boosts batch size by 60% without increasing memory consumption. For challenging tasks such as Math and Code Completion, LogQuant improves accuracy by 40% to 200% at the same compression ratio, outperforming comparable techniques. LogQuant integrates effortlessly with popular inference frameworks like Python's `transformers` library and will be made open-source upon publication.

## 1 Introduction

As Large Language Models (LLMs) continue to evolve, their capacity to process extended context lengths has increased significantly, from 4k to 128k tokens (Meta, 2024; OpenAI, 2024a). This improvement is particularly important for applications such as multi-round chatbot conversations (OpenAI, 2024a; Anthropic, 2024; DeepSeek, 2024) and document-based question answering (Gao et al., 2023; Lewis et al., 2020), where comprehensive contextual understanding is required. Moreover, the emergence of new models, such as OpenAI's o1 (OpenAI, 2024b), has increased the demand for even longer reasoning contexts, which exacerbates the memory challenges faced in KV cache management.

Recent works, such as Zhang et al. (2024); Li et al. (2024); Dong et al. (2024), have highlighted the significant memory consumption of the KV cache in large language models, which grows linearly with context length and can exceed the model's parameter size, presenting serious deployment challenges; a comparative analysis of these methods reveals their limitations in addressing memory efficiency, which our approach aims to overcome.

Various methods have been proposed to compress the KV cache, primarily focusing on either *eviction* or *quantization* strategies. Eviction-based approaches, such as H2O (Zhang et al., 2024), Keyformer (Adnan et al., 2024), StreamingLLM (Xiao et al., 2023), and snapKV (Li et al., 2024), aim to reduce memory usage by selectively removing tokens deemed unimportant. In contrast, quantization techniques, like QAQ (Dong et al., 2024), Gear (Kang et al., 2024), and KiVi (Liu et al., 2024c), reduce the precision of less important tokens, retaining more data while minimizing memory costs. Despite their differing approaches, both strategies face a common challenge: identifying which tokens are less important and, therefore, more suitable for compression. Methods such as KiVi and StreamingLLM address this by noting that tokens closer to the current position tend to be more important, so they focus on compressing or evicting tokens further from the current context. On the other hand, H2O predicts token importance based on attention scores from previous tokens.

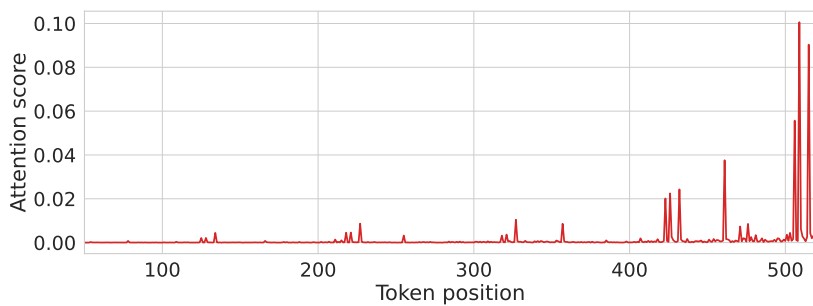

Figure 1: The observed log-distribution pattern is evident not only in the magnitude of attention scores but also in the positions of attention spikes. These spikes become sparser as the model attends to tokens further from the most recent position, indicating that the model not only focuses on nearby tokens. This phenomenon, illustrated here with Llama3-8B-Instruct (Dubey et al., 2024) on the GSM8K dataset (Cobbe et al., 2021), is consistent across different tasks and models, as further detailed in Section 3.

However, these methods introduce trade-offs: KiVi and StreamingLLM risk compressing important tokens outside their defined window, while H2O's reliance on past attention scores may lead to mispredictions, potentially reducing accuracy.

Our approach addresses these shortcomings by leveraging a key insight: the positions of the *attention spikes* (i.e. high attention scores) follow a log distribution as shown in Figure 1, resulting in sparser importance for tokens as they move further from the current position. By utilizing this property, we can outperform existing methods across a wide range of tasks. Additionally, the original absolute positions of KV cache entries can be disregarded without changing the final attention results during the decoding phase, which allows us to enhance the speed of our log-distributed quantization method.

The key contributions of this paper are as follows:

- **Observation of Log-Distributed Attention Spikes**: We observe that in various models and downstream tasks, the positions of high attention spikes follow a log distribution, becoming sparser as tokens move further from the current position. This insight underpins our approach to estimate token importance.

- **Design of LogQuant**: Leveraging this log-distribution observation, we introduce LogQuant, a 2-bit quantization technique that significantly improves accuracy. LogQuant outperforms existing methods like KiVi and H2O by better preserving important tokens, achieving a 40% to 200% improvement in accuracy on complex tasks such as Math and Code Completion with the same or higher compression ratio.

- **Throughput Optimization**: By ignoring the absolute positions of KV cache entries, our method further optimizes the speed of quantization/dequantization process without affecting the final attention results, resulting in a 25% increase in throughput and a 60% increase in batch size.

The remainder of the paper is organized as follows: Section 2 reviews the related work on KV cache compression techniques, Section 3 details the core concepts behind our proposed LogQuant methods, Section 4 present an extensive set of experiments, Section 5 summarizes our findings and discusses potential directions for future work.

## 2 BACKGROUND & RELATED WORK: KV CACHE COMPRESSION

In transformer models, the attention mechanism relies on three key components: the Query (Q), Key (K), and Value (V) vectors. For each token, the model computes a $d$-dimensional Query vector and compares it against all stored $N \times d$ Key vectors, where $N$ is the length of the sequence processed

so far. The result of this comparison is used to weigh the corresponding Value vectors, producing the final output. Mathematically, the attention operation is defined as:

$$\text{Attention}(Q, K, V) = \text{Softmax}\left(\frac{QK^\top}{\sqrt{d}}\right)V \tag{1}$$

Here, the Query vector is multiplied by the transposed Key matrix, resulting in a set of attention weights. These weights are then normalized using the softmax function, which reduces the $N$ sequence length dimension and are applied to the Value vectors to compute the output.

In existing literature, LLM inference is typically described in two phases: the prefill phase for processing input tokens and the decoding phase for generating new tokens. In the decoding phase, each token generation requires loading the entire KV Cache from previous tokens, leading to inefficiencies in both execution time and memory usage.

KV cache compression methods can be categorized into two distinct types: 'training-free' methods, which do not require model retraining and include eviction and quantization strategies, and 'training-required' methods, involve designing more efficient attention structures. Our approach focuses on improving training-free methods for broader applicability. Eviction methods discard less important tokens based on selective strategies, while quantization reduces the precision of key and value states to lower bits for memory efficiency. However, both methods face significant information loss at high compression rates—especially with 2-bit quantization, which can result in substantial accuracy degradation.

Inspired by attention patterns as Figure 1, we propose combining a logarithmic eviction strategies with quantization. By selectively retaining tokens in their original precision at critical positions during 2-bit quantization, we can preserve accuracy even at high compression rates.

## 2.1 KV Cache Eviction

Eviction methods aim to reduce KV cache memory usage in Large Language Models (LLMs) by discarding less important tokens. The early work H2O (Zhang et al., 2024) selects "heavy hitter" tokens based on cumulative attention scores, though this risks evicting tokens that may become important later. Keyformer (Adnan et al., 2024) improves on H2O by combining "Key Attention" with a "window attention" mechanism, retaining both historically significant and recent tokens for better accuracy. MiniCache (Liu et al., 2024b) reduces memory by reusing Key and Value states across layers. This method assumes that some key and value representations are redundant across model layers and can be shared. InfLLM (Xiao et al., 2024) addresses very long contexts by dividing them into blocks and retaining 'representative tokens' for block eviction decisions.

## 2.2 KV Cache Quantization

Quantization reduces storage and boosts computational speed by using fewer bits to represent values. Earlier works, like AWQ (Lin et al., 2023) and Qserve (Lin et al., 2024), applied 4-bit quantization to the KV cache with minimal accuracy loss. Recent methods aim to compress the KV cache further while preserving accuracy. QAQ (Dong et al., 2024) dynamically adjusts the precision of the in-GPU quantized cache by offloading all original-precision KV data to CPU memory. GEAR (Kang et al., 2024) improves accuracy by storing the quantization error of the KV cache as a sparse matrix with low-rank decomposition. KiVi (Liu et al., 2024c) introduces a 2-bit quantization by retaining a recent window of full-precision tokens, balancing memory efficiency and accuracy.

## 2.3 training-required approaches

An early memory-reducing attention design is Multi-Query Attention (MQA, (Shazeer, 2019)), where all query heads share a single pair of key and value heads. While this reduces memory, it significantly impacts accuracy. Grouped-Query Attention (GQA, (Ainslie et al., 2023)) addresses this by grouping query heads, with each group sharing the same key and value heads, preserving the generalization ability of multi-head attention while reducing KV cache size. Deepseek V2 (Liu et al., 2024a) introduces Multi-Head Latent Attention (MLA), which compresses key and value

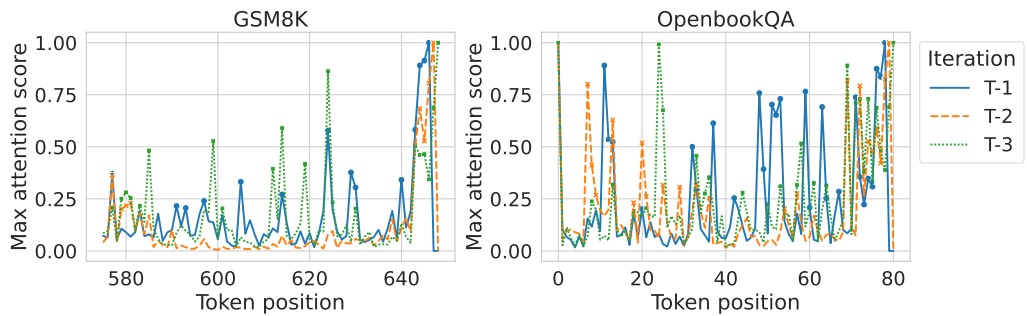

Figure 2: The maximum attention score of each token position across four consecutive decoding steps, marking the high attention positions for illustrating the unpredictable nature of attention scores. This analysis was conducted using Llama3-8B-Instruct (Dubey et al., 2024) on the GSM8K (Cobbe et al., 2021) and OpenBookQA (Mihaylov et al., 2018) datasets.

Table 1: Impact of retaining the first two tokens (referred to as "Sink") at original precision. The final answer accuracy results on GSM8K (Cobbe et al., 2021) are presented. We present the improvement as $\Delta_{\text{Sink}}$. Both methods maintain the recent 128 tokens at original precision.

| Model | baseline(BF16) | KiVi(4-bit) | KiVi(2-bit) | KiVi(2-bit)+Sink(BF16) | $\Delta_{Sink}$ |
|---|---|---|---|---|---|
| Llama3.1-8B-Instruct | 71.41 | 67.24 | 18.04 | 18.49 | +0.45 |
| Qwen1.5-7B-Chat | 57.24 | 52.27 | 39.80 | 39.42 | -0.38 |

states using LoRA-based projections. To prevent disruption of position embeddings from LoRA compression, specific channels are reserved for position information only, excluding them from LoRA compression.

## 3 METHODOLOGY

In Section 3.1, we explore the attention score distribution and analyze how quantization loss influences the attention block output. In Section 3.2, we present our observations on KV Cache and token importance. A position-agnostic attention calculation method is introduced in Section 3.3 for speeding up the log-distributed quantization method. Finally, we introduce the implementation of our **LogQuant** method in Section 3.4.

### 3.1 PRELIMINARY STUDY OF KV CACHE QUANTIZATION AND ATTENTION SCORES

As discussed in Section 2, two well-established observations in recent works are particularly relevant to KV cache compression. First, many tokens exhibit consistently low attention scores, indicating that their KV cache entries can be safely compressed with minimal impact on performance (Liu et al., 2024c). Second, predicting token importance based on previous decoding steps is unreliable, as attention scores can vary significantly across iterations, making it difficult to accurately identify which tokens should be preserved (Dong et al., 2024; Jiang et al., 2024). This is also demonstrated in Figure 2.

Inspired by the observation of *sink tokens* (Xiao et al., 2023), which are the first few tokens that consistently receive high attention scores (Figure 3), we included these tokens in the set maintained at original precision to improve accuracy in 2-bit quantization. However, as shown in Table 1, this adjustment yielded minimal improvement. This suggests that while sink tokens play a role in defining the conversational context, maintaining high precision for only these tokens is insufficient, indicating that tokens beyond the first few are also crucial for preserving model performance.

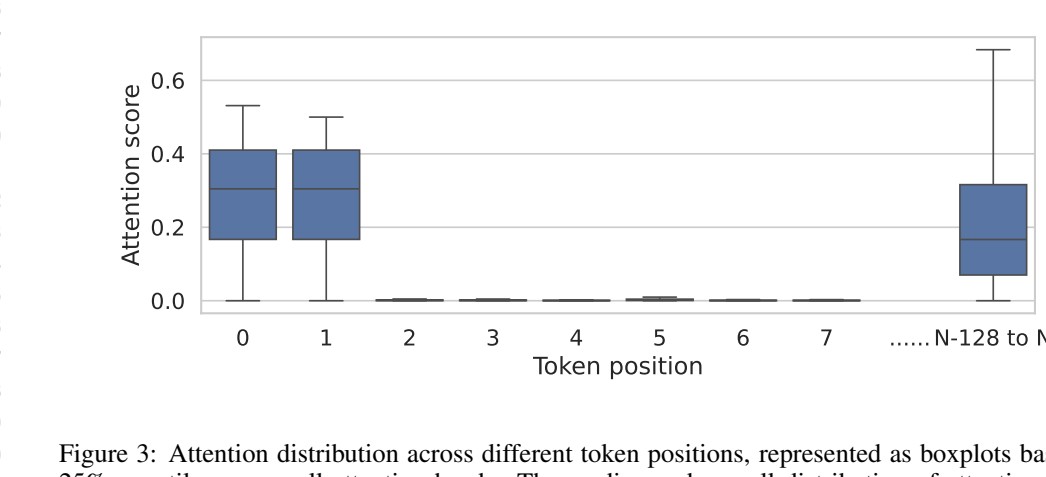

Figure 3: Attention distribution across different token positions, represented as boxplots based on 25% quantiles across all attention heads. The median and overall distribution of attention scores for sink tokens (Xiao et al., 2023) (tokens 0 and 1) are greater than the sum of the most recent 128 tokens. The attention scores are derived from experiments using Llama3-8B-Instruct (Dubey et al., 2024) and the GSM8K (Cobbe et al., 2021) dataset.

## 3.2 THE LOG-DISTRIBUTED ATTENTION PATTERN

As mentioned in Section 1, our analysis of attention heads reveals a log-distributed high-attention pattern, which motivates the development of a quantization scheme that follows this distribution. We introduce a selection scheme where a window of size $2W$ retains the most recent consecutive tokens in full precision. Following this, another window of size $W/2$ selects tokens spaced one token apart, and then a window of size $W/4$ follows the similar pattern and so on. Finally, a window of $3W$ tokens is reserved in full precision. This creates a log-distributed token selection scheme.

We compare this log-distributed selection to other methods: KiVi, which selects only the most recent $3W$ tokens; StreamingLLM, which selects the most recent $3W$ tokens plus the first four *sink tokens*; and H2O, which uses previous attention scores to select the top $3W$ tokens. To evaluate these methods, we define *token coverage* as the average attention score captured by the selection scheme:

$$\text{Token Coverage} = \frac{\sum_{i=1}^{3W} \text{Attention Score of Selected Tokens}}{3W}. \tag{2}$$

Figure 4 presents the results, where we exclude the first two tokens for calibration, as they typically have high attention scores but contribute minimally to overall model performance (see Section 3.1).

The results demonstrate that our log-distributed selection scheme covers high-attention tokens more effectively. This suggests that filtering tokens for quantization based on this log distribution leads to better token importance preservation.

## 3.3 POSITION-AGNOSTIC ATTENTION CALCULATION

LLM inference involves two phases: prefill and decoding (Section 2). As described in Yuan et al. (2024), the decoding phase is computationally expensive and memory-bound due to the use of the KV Cache. In the prefill phase, the model processes the input prompt in a single pass. However, during decoding, new tokens are generated one at a time, and each generation step requires access to the entire KV Cache. This leads to inefficiencies in both memory usage and execution time.

To mitigate these inefficiencies, we plan to accelerate the attention procedure. The attention operation can be expressed mathematically as follows:

$$A = \text{Softmax}(Q \cdot K^T)$$
$$O = A \cdot V, \tag{3}$$

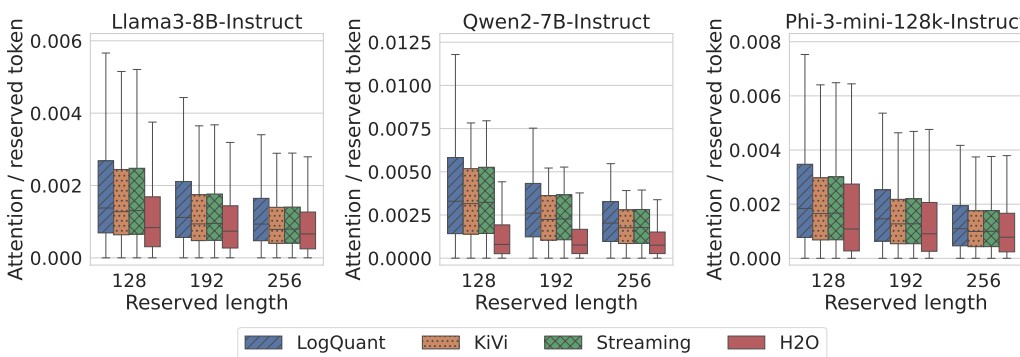

Figure 4: The attention coverage without the first two sink tokens for different selection methods (Liu et al., 2024c; Xiao et al., 2023; Zhang et al., 2024) and different models (Dubey et al., 2024; Yang et al., 2024; Abdin et al., 2024), tested on a subset of the GSM8K (Cobbe et al., 2021) dataset. Details of LogQuant will be introduced in Section 3.4.

where $A$ is the attention distribution, a $1 \times N$ vector resulting from the softmax operation applied to the product of $Q$ and the transpose of $K$ and $O$ is the output, a $1 \times d$ vector calculated by multiplying the attention distribution $A$ with the Value matrix $V$.

Since the attention distribution $A$ aggregates values over all $N$ tokens, the specific ordering of tokens in the Key and Value matrices does not affect the final output. This property allows us to permute or reorder the Key and Value caches without any loss of accuracy. By leveraging this insight, we can optimize the KV Cache by concatenating high-precision tokens with quantized tokens while disregarding their original positions. This approach enhances memory locality and processing efficiency while maintaining the correctness of the attention computation. This leads to the relation:

$$A \cdot V = A_P \cdot V_P, \tag{4}$$

where $P$ is a permutation of the indices $\{1, \ldots, N\}$. This enables us to optimize the KV Cache effectively.

### 3.4 LOGQUANT: ALGORITHM AND IMPLEMENTATION

**Algorithm.** After comparing different logarithmic bases $\log_N$, we found that a base-2 logarithmic implementation is sufficiently effective for our purposes. To maintain logarithmic sparsity within a specified length, we adopt this base-2 logarithmic approach. We fix a window length configuration $W$, allowing us to retain up to $3W$ tokens at original precision. Each time the length limit is reached, we reduce the density of tokens in the first two windows (each of length $W$) by retaining tokens at regular intervals, effectively halving the density. This process reduces the number of retained tokens in the first two windows from $2W$ to $\frac{2W}{2} = W$. Subsequently, we add $W$ new tokens, resulting in a full-precision window size of $\frac{2W}{2} + W = 2W$. At this point, the densities become $\text{density}_{W_1} = \frac{1}{2}p$ and $\text{density}_{W_2} = p$, where $p$ is the initial density and $W_i$ denotes the $i$-th window. By continuously adding new tokens, LogQuant naturally forms a $\log_2$ sparsity selection within the constrained length. The detailed selection process is described in Algorithm 1. Using this approach, the length of retained full-precision tokens fluctuates between $2W$ and $3W$, providing a more stable compression ratio compared to KiVi, where the length fluctuates between $0$ and $R$, with $R$ being the length of retained full-precision tokens in KiVi. We illustrate the workflow in Figure 5, which visually represents the KV cache management process, enhancing the understanding of our algorithm's implementation.

**Implementation.** Popular inference frameworks, such as Hugging Face's `transformers` library, have encapsulated KV Cache management into dedicated classes, which simplifies the integration of new methods. To leverage this modular design, we implemented **LogQuant** as a derived class of the `Cache` class in the `transformers` library. This approach ensures seamless compatibility with

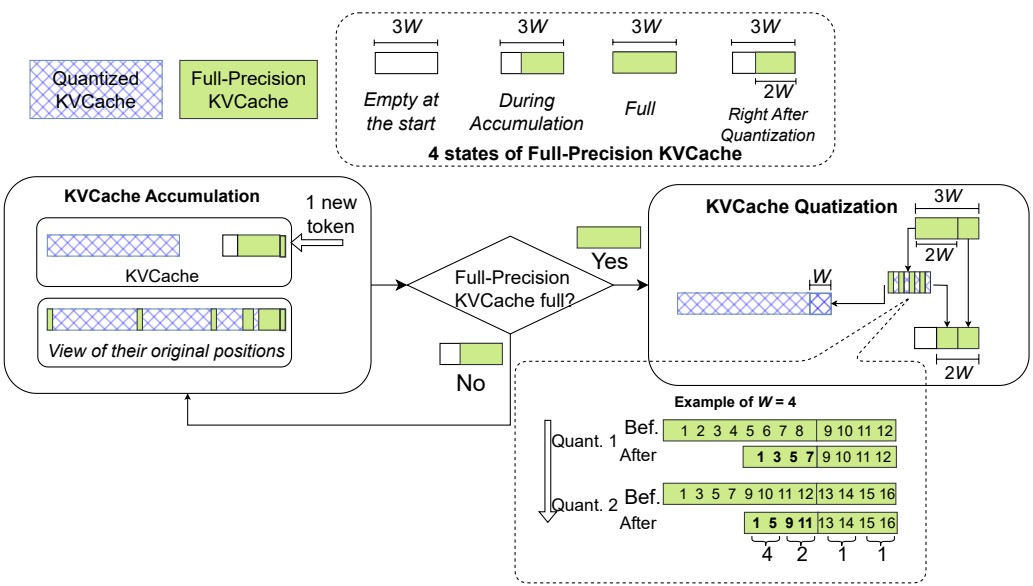

Figure 5: LogQuant's KV cache compression workflow. The number of reserved original-precision tokens increases from $2W$ to $3W$. We then apply a log-sparse strategy to filter the first $2W$ tokens, quantize half of these tokens, and compress the reserved token length back to $2W$.

---

**Algorithm 1** Log-based Filtering Token Selection Strategy

---

0: **Input:** $A$ (list of original precision tokens), $a*$ (new token), $W$ (window length)
0: **Output:** $A$ (updated list of tokens)
0: **procedure** APPENDTOKEN($A, a^*, W$)
0:     **if** length($A$) $< 3W$ **then**
0:       $A \leftarrow$ concat($A$, $a*$)
0:     **else**
0:       $A \leftarrow$ concat($A$[0:2W:2], $A$[2W:3W])
0:       $A \leftarrow$ concat($A$, $a*$)
0:     **end if**
0:     **return** $A$
0: **end procedure** =0

---

various quantization backends, including Quanto (Face, 2024) and HQQ (Badri & Shaji, 2023). For our implementation, we utilized Quanto as the quantization backend, adopting the Key-per-channel strategy. Furthermore, we integrated **LogQuant** into Hugging Face's inference pipeline, enhancing its usability for efficient and precise inference workflows.

Additionally, to assess the compression sensitivity of the Key and Value caches, we developed a variant called *PartialLogQuant*. This method log-sparsely selects original precision tokens exclusively for the Key cache while reserving only the most recent $W$ tokens for the Value cache.

## 4 EXPERIMENTS

### 4.1 SETTINGS

**Models.** We evaluate KiVi and *LogQuant* by 3 popular model families: Llama3/Llama3.1 (Dubey et al., 2024), Qwen1.5/Qwen2 (Bai et al., 2023; Yang et al., 2024), and Microsoft Phi3 (Abdin et al., 2024). Qwen1.5 and Phi3 are based on Multi-Head Attention, whereas Llama3/3.1 and Qwen2 utilize Group-Query Attention. The quantization group size $G$ is set to the Hugging Face default value of 64, and the quantized precision is set to INT2. For KiVi, the maximum length of reserved original-precision tokens $R$ is set to [128, 192, 256]. For LogQuant, the window length $W$ is limited

to $\lfloor \frac{R}{3} \rfloor$ as it will reserve a maximum of $3W$ original precision tokens and for PartialLogQuant, which reserve $3W$ Key cache and $W$ Value cache in original precision, we set $W = \lfloor \frac{R}{2} \rfloor$ to ensure that the total number of reserved original-precision tokens does not exceed that of KiVi.

**Datasets.** We selected GSM8K(Grade School Math, (Cobbe et al., 2021)) and LongBench (Bai et al., 2024) due to their widespread use in evaluating KV cache quantization, ensuring our results are comparable to those in the literature. For GSM8K, we test with a 5-shot from the training set for better accuracy and keep the length of the input token between 600 and 1700, the evaluation is based on the exact value of the final answer. For LongBench, we test all 21 datasets among 6 types of tasks and use the LongBench's original pipeline for evaluation. The test dataset details are present in Table 5.

### 4.2 ACCURACY AND EFFICIENCY ANALYSIS

#### 4.2.1 ACCURACY COMPARISON ON DIFFERENT PRECISION

To illustrate the impact of quantized data precision, we evaluate the accuracy loss using Llama3.1-8B-Instruct under both 2-bit and 4-bit quantization for KiVi and LogQuant methods on LongBench. As shown in Table 2, both methods achieve performance comparable to the baseline across all tasks with 4-bit quantization. However, 2-bit quantization results in a noticeable drop in accuracy, highlighting the trade-off between memory efficiency and performance. Notably, LogQuant demonstrates better accuracy compared to KiVi under the same conditions.

Table 2: Accuracy of Different Precision on Llama3.1-8B. Refer to the Table 7 for the scores of each specific task. The $\Delta$ shows the difference to baseline.

| Category | KiVi (2-bit) | KiVi (4-bit) | LogQuant (2-bit) | LogQuant (4-bit) | baseline |
|---|---|---|---|---|---|
| Single-Document QA | $38.89\,(\Delta - 8.11)$ | $47.75\,(\Delta + 0.75)$ | $41.91\,(\Delta - 5.09)$ | $47.73\,(\Delta + 0.73)$ | 47.71 |
| Multi-Document QA | $34.02\,(\Delta - 4.98)$ | $39.74\,(\Delta + 0.74)$ | $36.08\,(\Delta - 2.92)$ | $39.93\,(\Delta + 0.93)$ | 39.96 |
| Summarization | $16.10\,(\Delta - 1.90)$ | $17.94\,(\Delta - 0.06)$ | $16.62\,(\Delta - 1.38)$ | $17.92\,(\Delta - 0.08)$ | 18.08 |
| Few-shot Learning | $52.51\,(\Delta - 8.49)$ | $61.34\,(\Delta + 0.34)$ | $56.43\,(\Delta - 4.57)$ | $61.21\,(\Delta + 0.21)$ | 61.22 |
| Synthetic Tasks | $45.02\,(\Delta - 21.98)$ | $67.74\,(\Delta + 0.74)$ | $52.51\,(\Delta - 14.49)$ | $67.68\,(\Delta + 0.68)$ | 67.78 |
| Code Completion | $43.06\,(\Delta - 15.94)$ | $59.53\,(\Delta + 0.53)$ | $52.10\,(\Delta - 6.90)$ | $59.57\,(\Delta + 0.57)$ | 59.78 |

#### 4.2.2 ACCURACY COMPARISON AMONG DIFFERENT CONFIGURATIONS

As discussed in Section 4.2.1, 4-bit quantization incurs only a slight accuracy loss across tasks. Therefore, we focus on 2-bit quantization in the following discussion to highlight LogQuant's performance. To further investigate the accuracy loss resulting from quantization, we compared the following methods: 1) 16-bit baseline, 2) KiVi, 3) LogQuant, and 4) PartialLogQuant across different configurations, we define the *compression ratio* as:

$$\frac{\text{Original tensor size}}{\text{Tensor size in compressed format}} \tag{5}$$

where, for a sequence length $L$ and reserved original precision token length $R$ in a BF16 model with 2-bit quantization, the *compression ratio* can be expressed as:

$$\frac{16L}{2(L - R) + 16R}. \tag{6}$$

We tested the three compression ratios using GSM8K across three model families, and the results summarized in Figure 6. Our findings demonstrate that the *LogQuant* method consistently outperforms KiVi across all three models at various compression ratios. Furthermore, at higher compression ratios, *PartialLogQuant* exhibits superior performance compared to standard *LogQuant*, which show a speculation that Key, the component for computing attention are more sensitive for quantization loss. The results also indicate that smaller models and small KV states models, such as Phi3-mini (3.8B) and Qwen2-7B (retaining only $\frac{1}{8}$ of KV heads than Query, while other GQA models typically retain at least $\frac{1}{4}$.), experience a more significant accuracy loss with 2-bit quantized KV caches. However, our method provides a notable improvement in accuracy for these smaller models.

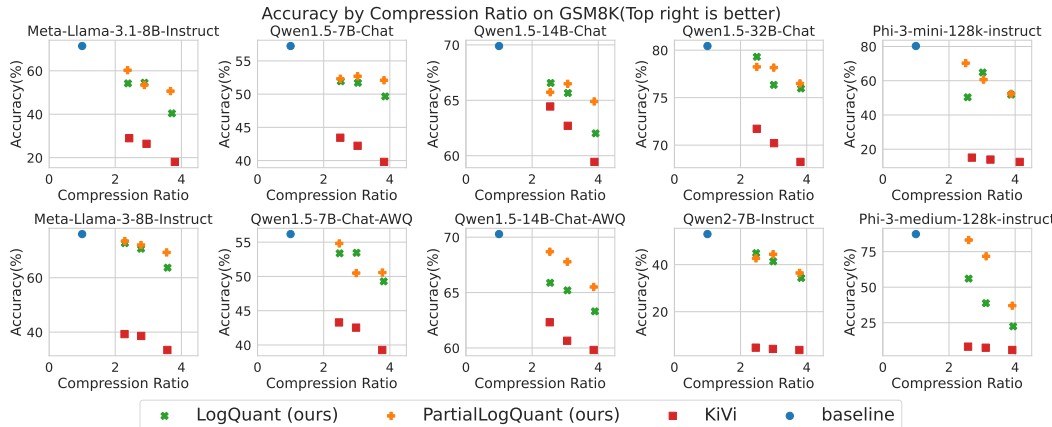

Figure 6: Accuracy(EM) with different compression ratio in GSM8K tasks for different models.

### 4.2.3 ACCURACY COMPARISON AMONG DIFFERENT TASKS

To further investigate the accuracy loss in different tasks, we evaluate the seven task groups listed in Table 5, providing the average score for each method in Table 3. We set the reserved length $R$ as 128, where LogQuant will have only $3\lfloor\frac{R}{3}\rfloor = 126$ original precision tokens, slightly smaller than 128 of KiVi. As shown in Table 3, for simpler tasks such as summarization, quantization has little to no impact on performance compared to the 16-bit baseline. However, for more complex tasks like Code Completion, Synthetic Tasks and Math, quantization significantly affects accuracy, with *LogQuant* demonstrating better retention of accuracy compared to KiVi.

### 4.2.4 EFFICIENCY COMPARISON

To evaluate memory and throughput efficiency by a NVIDIA H100 48G MIG with the HuggingFace pipeline, we conducted a benchmark similar to that in (Turganbay, 2024), setting an average prompt length of 512 and a maximum output length of 2000. We incrementally increased the batch size while recording peak memory usage and throughput for both *LogQuant* (2-bit with 126 reserved tokens) and the BF16 baseline on the Llama-3.1-8B model, until memory usage reached the 48GB limit. The hardware utilized was a single NVIDIA H100 GPU. As shown in Figure 7, *LogQuant* achieves approximately 25% higher throughput by supporting a larger batch size. Additionally, it allows for a 60% increase in batch size within the same memory constraints under the HuggingFace pipeline.

We also observed that, within the HuggingFace pipeline, inference with a quantized cache does not immediately release original KV states, which limits memory compression and efficiency. Furthermore, the dequantization operation impacts throughput. These issues suggest that memory efficiency and speed could be further improved by employing operator fusion, enabling computation on the quantized cache directly with a fused attention operation. We will explore this optimization in future work.

## 5 CONCLUSION AND FUTURE WORK

In this paper, we introduced LogQuant, a novel quantization technique designed to optimize KV Cache management in large language models (LLMs). Our approach leverages a base-2 logarithmic strategy to maintain sparsity while accommodating an increased number of full-precision tokens. Through comprehensive evaluations, we demonstrated that LogQuant consistently outperforms existing methods, such as KiVi, across various model families and compression ratios, particularly benefiting smaller models that typically suffer from accuracy loss due to quantization.

We further explored the efficiency of our implementation within the HuggingFace pipeline, achieving notable improvements in throughput and memory utilization. Additionally, our investigation into accuracy loss across different tasks highlighted LogQuant's superior retention of performance,

Table 3: Task Group Average Score for Different Models and Methods.
(The best result of 2-bit quantization will be bold. Refer to the Table 6 for the scores of each specific task in LongBench)

| Model | precision Task Group | 16-bit Baseline | KiVi | 2-bit LogQuant (ours) | PartialLogQuant (ours) |
|---|---|---|---|---|---|
| llama-3.1-8B-Instruct | Math | 71.42 | 18.04 | 40.41 | **50.64** |
| | Code Completion | 59.78 | 43.06 | 52.09 | **52.36** |
| | Few-shot Learning | 61.21 | 52.50 | 56.42 | **56.91** |
| | Multi-Document QA | 39.95 | 34.01 | **36.08** | 35.80 |
| | Single-Document QA | 47.71 | 38.89 | 41.90 | **42.48** |
| | Summarization | 18.07 | 16.10 | 16.62 | **16.74** |
| | Synthetic Tasks | 67.78 | 45.02 | **52.51** | 52.11 |
| Qwen1.5-7B-Chat-AWQ | Math | 56.18 | 39.27 | 49.28 | **50.57** |
| | Code Completion | 52.46 | 34.79 | 40.68 | **43.11** |
| | Few-shot Learning | 53.88 | 51.32 | **52.54** | 52.46 |
| | Multi-Document QA | 33.05 | 31.08 | **32.04** | 31.80 |
| | Single-Document QA | 39.26 | 35.80 | 37.22 | **37.3** |
| | Summarization | 17.11 | 17.16 | **17.38** | 17.31 |
| | Synthetic Tasks | 26.5 | 10 | 13.5 | **13.66** |
| Qwen1.5-14B-Chat-AWQ | Math | 70.28 | 59.82 | 63.31 | **65.50** |
| | Code Completion | 57.47 | 37.48 | 49.37 | **50.44** |
| | Few-shot Learning | 59.02 | 57.50 | **58.25** | 58.22 |
| | Multi-Document QA | 39.72 | 37.91 | 38.01 | **38.14** |
| | Single-Document QA | 42.48 | 40.39 | **41.37** | 41.31 |
| | Summarization | 17.21 | 17.17 | **17.24** | 17.21 |
| | Synthetic Tasks | 61.33 | 46.85 | **52.17** | 52.00 |
| Qwen2-7B-Instruct | Math | 52.99 | 3.71 | 34.34 | **36.47** |
| | Code Completion | 58.23 | 35.91 | 48.71 | **49.56** |
| | Few-shot Learning | 61.90 | 35.26 | **51.23** | 51.04 |
| | Multi-Document QA | 33.35 | 12.35 | **28.28** | 28.19 |
| | Single-Document QA | 44.66 | 20.52 | 34.84 | **35.46** |
| | Summarization | 16.33 | 9.31 | 13.13 | **13.34** |
| | Synthetic Tasks | 43.00 | 11.42 | 22.83 | **24.17** |
| Phi-3-mini-128k-instruct | Math | 80.29 | 12.59 | 51.86 | **52.39** |
| | Code Completion | 55.97 | 33.97 | **40.84** | 40.33 |
| | Few-shot Learning | 52.58 | 36.17 | 39.36 | **40.07** |
| | Multi-Document QA | 33.55 | 18.19 | 21.70 | **22.05** |
| | Single-Document QA | 42.47 | 19.58 | **23.63** | 23.63 |
| | Summarization | 17.56 | 9.10 | 9.89 | **10.30** |
| | Synthetic Tasks | 48.00 | 4.83 | 5.39 | **6.15** |

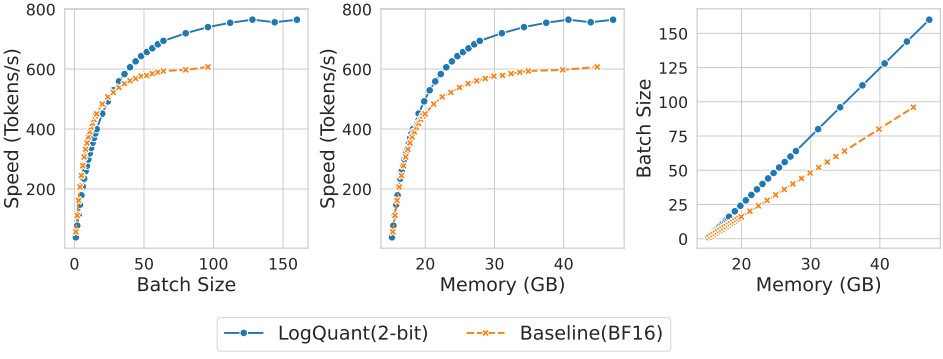

Figure 7: memory usage and throughput comparison between 2bit LogQuant and 16bit baseline under huggingface generation pipeline with llama3.1-8B and H100.

especially in complex tasks. These findings underscore the potential of LogQuant to enhance LLM inference in resource-constrained environments.

Future work will focus on refining our quantization approach and investigating further optimizations, such as operator fusion, to maximize efficiency and performance in LLM applications.

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

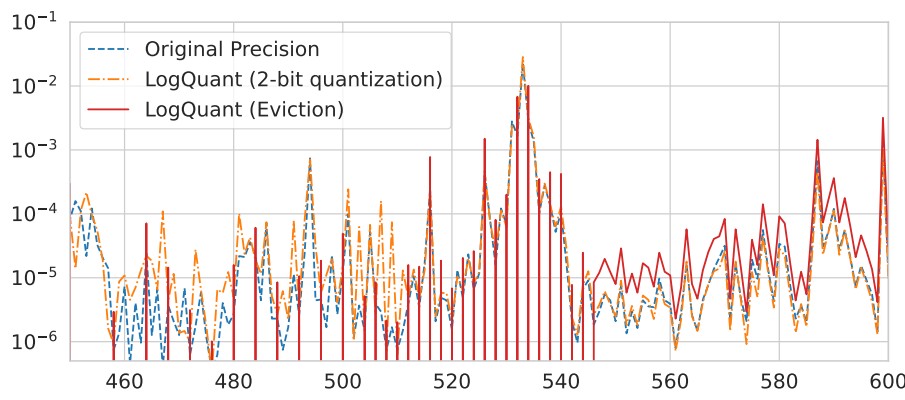

Figure 8: Eviction and Quantization Loss on Attention Distribution

## A  DISCUSSION ON WHY NOT EVICTION

Unlike quantization, which only impacts the precision of specific tokens, eviction alters the sequence length directly. Attention is computed using the softmax function, which scales all values to sum to 1. Due to this property, eviction methods can result in much larger deviations from the baseline compared to quantization within the fully preserved window. Furthermore, for the dropped segments, eviction methods are unable to compute attention, leading to significantly higher errors.

We illustrate this behavior in Figure 8 and summarize the attention error relative to the baseline for Llama3.1-8B on the GSM8K dataset in Table 4.

Table 4: Comparison of L1 error with original attention for eviction and quantization.

| LogQuant (2-bit) | KiVi (2-bit) | LogQuant (Eviction) | KiVi (Eviction) |
|---|---|---|---|
| 432.50 | 556.10 | 1076.70 | 1612.56 |

## B  OVERVIEW OF TEST DATASETS

## C  META DATA OF LONGBENCH RESULTS

Table 6: LongBench score of each dataset

| precision | 16-bit | 2-bit | | |
|---|---|---|---|---|
| Task Group | Baseline | KiVi | LogQuant (ours) | PartialLogQuant (ours) |
| | | llama-3-8B-Instruct | | |
| 2WikiMultihopQA | 37.24 | 31.72 | 35.08 | **35.79** |
| DuReader | 16.73 | 12.45 | 15.5 | **15.69** |
| GovReport | 17.8 | 12.8 | 15.63 | **16.37** |
| HotpotQA | 46.1 | 43.87 | **44.96** | 44.73 |
| LCC | 56.85 | 31.73 | 41.75 | **44.61** |
| LSHT | 25.25 | 21.5 | **21.75** | 21.75 |
| MultiFieldQA-en | 44.44 | 38.68 | 41.04 | **41.68** |
| MultiFieldQA-zh | 56.3 | 43.96 | 48.44 | **48.64** |
| MultiNews | 16.59 | 15.76 | **16.06** | 15.79 |
| MuSiQue | 21.44 | 19.56 | **20.59** | 20.56 |
| NarrativeQA | 22.07 | 19.82 | 21.56 | **21.81** |
| PassageCount | 6.5 | **5.5** | 4.0 | 5.0 |

**Table 6 – continued from previous page**

| Task Group | Baseline | KiVi | LogQuant (ours) | PartialLogQuant (ours) |
|---|---|---|---|---|
| PassageRetrieval-en | 66.0 | 53.0 | 58.5 | **59.0** |
| PassageRetrieval-zh | 91.0 | 33.45 | 72.0 | **72.5** |
| Qasper | 43.69 | 33.9 | **39.46** | 39.38 |
| QMSum | 17.49 | 17.01 | 17.37 | **17.48** |
| RepoBench-P | 51.32 | 31.99 | 40.1 | **41.59** |
| SAMSum | 33.22 | 22.44 | 32.66 | **33.15** |
| TREC | 74.0 | 72.5 | **73.0** | 73.0 |
| TriviaQA | 90.48 | 87.65 | **89.36** | 88.59 |
| VCSUM | 0.16 | 0.17 | **0.25** | 0.2 |
| **llama-3.1-8B-Instruct** | | | | |
| 2WikiMultihopQA | 45.06 | 39.52 | **40.69** | 39.61 |
| DuReader | 28.48 | 22.2 | 22.59 | **22.63** |
| GovReport | 20.41 | 18.6 | 18.78 | **18.96** |
| HotpotQA | 55.9 | 48.83 | **52.43** | 52.06 |
| LCC | 62.99 | 47.09 | 57.52 | **57.55** |
| LSHT | 45.0 | 31.42 | 33.75 | **34.0** |
| MultiFieldQA-en | 54.91 | 42.51 | 45.98 | **47.17** |
| MultiFieldQA-zh | 62.72 | 50.12 | 55.51 | **55.57** |
| MultiNews | 15.89 | 15.07 | 15.11 | **15.28** |
| MuSiQue | 30.39 | 25.52 | 28.62 | **28.93** |
| NarrativeQA | 28.19 | 26.44 | 27.93 | **28.17** |
| PassageCount | 6.31 | **5.67** | 5.63 | 5.63 |
| PassageRetrieval-en | 99.5 | 83.17 | **92.25** | 91.5 |
| PassageRetrieval-zh | 97.54 | 46.23 | **59.65** | 59.2 |
| Qasper | 45.03 | 36.5 | 38.21 | **39.01** |
| QMSum | 19.15 | 17.41 | 18.19 | **18.2** |
| RepoBench-P | 56.57 | 39.03 | 46.67 | **47.18** |
| SAMSum | 35.72 | 23.88 | 33.33 | **34.26** |
| TREC | 72.5 | 65.0 | 67.0 | **68.0** |
| TriviaQA | 91.64 | 89.72 | **91.63** | 91.41 |
| VCSUM | 16.85 | 13.33 | 14.41 | **14.52** |
| **Phi-3-mini-128k-instruct** | | | | |
| 2WikiMultihopQA | 35.78 | 19.12 | 24.61 | **24.96** |
| DuReader | 22.75 | **10.38** | 9.26 | 8.66 |
| GovReport | 18.7 | 8.83 | 9.47 | **9.96** |
| HotpotQA | 50.44 | 31.33 | 37.48 | **38.66** |
| LCC | 57.44 | 39.85 | **47.53** | 47.41 |
| LSHT | 27.25 | 14.25 | 13.75 | **14.75** |
| MultiFieldQA-en | 54.9 | 29.04 | **34.91** | 33.71 |
| MultiFieldQA-zh | 52.09 | 8.16 | **12.32** | 11.87 |
| MultiNews | 15.52 | 12.72 | 13.33 | **13.36** |
| MuSiQue | 25.23 | 11.92 | 15.46 | **15.93** |
| NarrativeQA | 23.28 | 15.34 | 17.37 | **18.26** |
| PassageCount | 3.0 | 2.25 | **4.5** | 3.0 |
| PassageRetrieval-en | 82.5 | 11.0 | 9.68 | **13.96** |
| PassageRetrieval-zh | 58.5 | 1.25 | **2.0** | 1.5 |
| Qasper | 39.6 | 25.78 | 29.91 | **30.68** |
| QMSum | 17.97 | 5.88 | 7.04 | **8.37** |
| RepoBench-P | 54.49 | 28.09 | **34.16** | 33.25 |
| SAMSum | 30.62 | 9.23 | 13.03 | **13.42** |
| TREC | 66.0 | 59.5 | **62.5** | 62.5 |
| TriviaQA | 86.43 | 61.72 | 68.15 | **69.6** |
| VCSUM | 18.04 | 8.97 | **9.74** | 9.5 |
| **Qwen1.5-14B-Chat-AWQ** | | | | |
| 2WikiMultihopQA | 44.81 | 44.35 | **44.39** | 44.39 |
| DuReader | 26.02 | 23.34 | 23.28 | **23.6** |

**Table 6 – continued from previous page**

| Task Group | Baseline | KiVi | LogQuant (ours) | PartialLogQuant (ours) |
|---|---|---|---|---|
| GovReport | 16.31 | 16.23 | 16.25 | **16.29** |
| HotpotQA | 55.67 | 53.69 | 53.9 | **53.95** |
| LCC | 56.69 | 36.94 | 50.95 | **51.78** |
| LSHT | 37.0 | 32.5 | **34.5** | 34.5 |
| MultiFieldQA-en | 48.36 | 44.75 | 45.68 | **45.69** |
| MultiFieldQA-zh | 60.35 | 58.54 | 59.43 | **59.44** |
| MultiNews | 14.95 | **15.01** | 14.94 | 14.94 |
| MuSiQue | 32.38 | 30.25 | 30.45 | **30.6** |
| NarrativeQA | 22.26 | 21.73 | **22.83** | 22.59 |
| PassageCount | 1.0 | **2.55** | 2.0 | 2.5 |
| PassageRetrieval-en | 94.5 | 71.0 | **80.0** | 79.0 |
| PassageRetrieval-zh | 88.5 | 67.0 | **74.5** | 74.5 |
| Qasper | 38.93 | 36.56 | **37.54** | 37.53 |
| QMSum | 18.16 | 18.03 | **18.13** | 18.09 |
| RepoBench-P | 58.25 | 38.03 | 47.79 | **49.1** |
| SAMSum | 32.95 | 32.69 | **33.34** | 32.86 |
| TREC | 77.5 | 76.5 | **77.5** | 77.5 |
| TriviaQA | 88.63 | **88.32** | 87.66 | 88.01 |
| VCSUM | 19.41 | 19.42 | **19.65** | 19.54 |
| **Qwen1.5-7B-Chat** | | | | |
| 2WikiMultihopQA | 32.8 | 31.83 | 32.14 | **32.53** |
| DuReader | 25.96 | 22.64 | **24.06** | 23.72 |
| GovReport | 16.66 | 15.57 | **15.84** | 15.83 |
| HotpotQA | 48.11 | 47.37 | **48.91** | 48.11 |
| LCC | 58.17 | 45.87 | 53.77 | **53.93** |
| LSHT | 28.0 | 24.0 | 24.5 | **25.0** |
| MultiFieldQA-en | 47.14 | 42.26 | 43.72 | **44.08** |
| MultiFieldQA-zh | 53.4 | 50.18 | **51.68** | 51.13 |
| MultiNews | 15.02 | **15.0** | 14.92 | 14.83 |
| MuSiQue | 26.74 | 25.88 | **27.09** | 26.33 |
| NarrativeQA | 20.06 | 19.02 | 20.06 | **20.5** |
| PassageCount | 1.0 | **0.5** | 0.0 | 0.5 |
| PassageRetrieval-en | 40.5 | 20.0 | 24.0 | **24.5** |
| PassageRetrieval-zh | 59.0 | 18.25 | **29.0** | 27.5 |
| Qasper | 39.84 | 37.19 | **37.28** | 37.13 |
| QMSum | 18.25 | 17.59 | **18.18** | 17.82 |
| RepoBench-P | 45.46 | 26.33 | 30.76 | **32.55** |
| SAMSum | 33.01 | 29.7 | **33.31** | 32.62 |
| TREC | 70.5 | **69.5** | 67.5 | 67.0 |
| TriviaQA | 86.76 | 86.51 | 87.37 | **87.79** |
| VCSUM | 17.98 | 19.15 | **19.34** | 19.26 |
| **Qwen1.5-7B-Chat-AWQ** | | | | |
| 2WikiMultihopQA | 32.43 | 30.82 | **33.46** | 32.94 |
| DuReader | 25.84 | 23.1 | **24.36** | 24.06 |
| GovReport | 16.98 | 16.31 | 16.65 | **16.7** |
| HotpotQA | 47.77 | **47.17** | 46.0 | 46.33 |
| LCC | 57.98 | 44.56 | 52.33 | **54.32** |
| LSHT | 29.0 | 25.5 | **27.0** | 27.0 |
| MultiFieldQA-en | 46.72 | 42.87 | 45.85 | **45.93** |
| MultiFieldQA-zh | 50.97 | 45.51 | 46.73 | **47.13** |
| MultiNews | 14.97 | 15.04 | **15.16** | 15.08 |
| MuSiQue | 26.18 | 23.23 | **24.36** | 23.9 |
| NarrativeQA | 20.93 | 19.58 | **20.14** | 19.94 |
| PassageCount | 0.5 | **0.0** | 0.0 | 0.0 |
| PassageRetrieval-en | 30.5 | 16.0 | **18.5** | 17.0 |
| PassageRetrieval-zh | 48.5 | 14.0 | 22.0 | **24.0** |

Continued on next page

**Table 6 – continued from previous page**

| Task Group | Baseline | KiVi | LogQuant (ours) | PartialLogQuant (ours) |
|---|---|---|---|---|
| Qasper | 38.45 | 35.27 | 36.16 | **36.2** |
| QMSum | 17.85 | 17.34 | **17.77** | 17.58 |
| RepoBench-P | 46.95 | 25.02 | 29.03 | **31.91** |
| SAMSum | 31.98 | 28.3 | **32.06** | 31.39 |
| TREC | 67.0 | **65.0** | 63.5 | 64.0 |
| TriviaQA | 87.56 | 86.48 | **87.61** | 87.48 |
| VCSUM | 18.66 | 19.95 | **19.96** | 19.91 |
| **Qwen2-7B-Instruct** | | | | |
| 2WikiMultihopQA | 44.15 | 11.33 | **40.12** | 40.02 |
| DuReader | 19.22 | 13.08 | **15.01** | 14.54 |
| GovReport | 18.09 | 10.82 | 16.07 | **16.74** |
| HotpotQA | 44.3 | 17.39 | **39.92** | 39.66 |
| LCC | 57.72 | 36.63 | 51.46 | **51.92** |
| LSHT | 44.0 | 23.0 | 26.25 | **28.25** |
| MultiFieldQA-en | 46.89 | 21.97 | 36.42 | **37.69** |
| MultiFieldQA-zh | 61.48 | 33.67 | **47.57** | 47.01 |
| MultiNews | 15.58 | 8.53 | 13.6 | **13.71** |
| MuSiQue | 25.71 | 7.58 | 18.07 | **18.53** |
| NarrativeQA | 24.43 | 5.29 | 18.43 | **18.56** |
| PassageCount | 5.0 | 5.5 | 5.5 | **6.0** |
| PassageRetrieval-en | 69.0 | 19.25 | 33.5 | **36.0** |
| PassageRetrieval-zh | 55.0 | 9.5 | 29.5 | **30.5** |
| Qasper | 45.82 | 21.16 | 36.94 | **38.58** |
| QMSum | 17.92 | 9.08 | **12.25** | 12.14 |
| RepoBench-P | 58.74 | 35.18 | 45.95 | **47.19** |
| SAMSum | 35.94 | 18.23 | **28.03** | 26.77 |
| TREC | 78.0 | 58.25 | **68.0** | 68.0 |
| TriviaQA | 89.66 | 41.56 | **82.63** | 81.15 |
| VCSUM | 13.74 | 8.82 | 10.58 | **10.77** |

Table 5: Overview of all test datasets.
'Avg len' (average length) is computed using the number of words for the English (code) datasets and the number of characters for the Chinese datasets. 'Accuracy (CLS)' refers to classification accuracy, while 'Accuracy (EM)' refers to exact match accuracy

| Task Group | Dataset | Avg len | Metric | Language | #data |
|---|---|---|---|---|---|
| **Math** | GSM8K | 240 | Accuracy (EM) | English | 1319 |
| **Single-Document QA** | NarrativeQA | 18,409 | F1 | English | 200 |
| | Qasper | 3,619 | F1 | English | 200 |
| | MultiFieldQA-en | 4,559 | F1 | English | 150 |
| | MultiFieldQA-zh | 6,701 | F1 | Chinese | 200 |
| **Multi-Document QA** | HotpotQA | 9,151 | F1 | English | 200 |
| | 2WikiMultihopQA | 4,887 | F1 | English | 200 |
| | MuSiQue | 11,214 | F1 | English | 200 |
| | DuReader | 15,768 | Rouge-L | Chinese | 200 |
| **Summarization** | GovReport | 8,734 | Rouge-L | English | 200 |
| | QMSum | 10,614 | Rouge-L | English | 200 |
| | MultiNews | 2,113 | Rouge-L | English | 200 |
| | VCSUM | 15,380 | Rouge-L | Chinese | 200 |
| **Few-shot Learning** | TREC | 5,177 | Accuracy (CLS) | English | 200 |
| | TriviaQA | 8,209 | F1 | English | 200 |
| | SAMSum | 6,258 | Rouge-L | English | 200 |
| | LSHT | 22,337 | Accuracy (CLS) | Chinese | 200 |
| **Synthetic Task** | PassageCount | 11,141 | Accuracy (EM) | English | 200 |
| | PassageRetrieval-en | 9,289 | Accuracy (EM) | English | 200 |
| | PassageRetrieval-zh | 6,745 | Accuracy (EM) | Chinese | 200 |
| **Code Completion** | LCC | 1,235 | Edit Sim | Python/C#/Java | 500 |
| | RepoBench-P | 4,206 | Edit Sim | Python/Java | 500 |

Table 7: Comparison on Llama3.1-8B-Instruct of different quantization precisions

| Dataset | KiVi (2-bit) | KiVi (4-bit) | LogQuant (2-bit) | LogQuant (4-bit) | Baseline |
|---|---|---|---|---|---|
| 2wikimqa | 39.52 | 44.79 | 40.69 | 45.18 | 45.06 |
| dureader | 22.20 | 27.75 | 22.59 | 27.99 | 28.48 |
| gov_report | 18.60 | 19.86 | 18.78 | 20.09 | 20.41 |
| hotpotqa | 48.83 | 55.78 | 52.43 | 55.85 | 55.90 |
| lcc | 47.09 | 63.44 | 57.52 | 62.85 | 62.99 |
| lsht | 31.42 | 45.00 | 33.75 | 45.00 | 45.00 |
| multi_news | 15.07 | 15.65 | 15.11 | 15.64 | 15.89 |
| multifieldqa_en | 42.51 | 55.10 | 45.98 | 54.63 | 54.91 |
| multifieldqa_zh | 50.12 | 62.77 | 55.51 | 63.27 | 62.72 |
| musique | 25.52 | 30.65 | 28.62 | 30.70 | 30.39 |
| narrativeqa | 26.44 | 27.91 | 27.93 | 28.28 | 28.19 |
| passage_count | 5.67 | 6.31 | 5.63 | 6.15 | 6.31 |
| passage_retrieval_en | 83.17 | 99.50 | 92.25 | 99.50 | 99.50 |
| passage_retrieval_zh | 46.23 | 97.42 | 59.65 | 97.38 | 97.54 |
| qasper | 36.50 | 45.20 | 38.21 | 44.74 | 45.03 |
| qmsum | 17.41 | 19.07 | 18.19 | 18.92 | 19.15 |
| repobench-p | 39.03 | 55.61 | 46.67 | 56.28 | 56.57 |
| samsum | 23.88 | 36.12 | 33.33 | 35.45 | 35.72 |
| trec | 65.00 | 72.50 | 67.00 | 72.50 | 72.50 |
| triviaqa | 89.72 | 91.73 | 91.63 | 91.89 | 91.64 |
| vcsum | 13.33 | 17.17 | 14.41 | 17.04 | 16.85 |

