# OpenReview forum: "LogQuant: Log-Distributed 2-Bit Quantization of KV Cache with Superior Accuracy Preservation"
_ICLR.cc/2025/Conference — Submitted to ICLR 2025_

### Official Review · Reviewer_JWgv · 2024-10-23

**Soundness:** 3
**Presentation:** 4
**Contribution:** 3
**Rating:** 6
**Confidence:** 4

**Summary:**

This paper 1) observes the log-distribution patterns of attention score magnitudes and attention spike locations, 2) introduces LogQuant to exploit these properties, and 3) shows that their approach can achieve higher accuracy and competitive throughput at the same compression ratio. LogQuant achieves this by applying a log-based filtering mechanism in the 2-bit quantization of the KV cache.

**Strengths:**

1. This paper observes an interesting phenomenon in the attention score pattern and designs a simple but effective KV cache quantization framework around it.
2. The paper is well-written.

**Weaknesses:**

1. Latency is the most important metric of efficiency, but it was not compared in the experiments.
2. The efficiency comparison does not use real-world inference traces and only includes a naive BF16 baseline.
3. The accuracy comparison only includes quantization baseline KiVi but no eviction-based baseline.

**Questions:**

Given that LogQuant and PartialLogQuant excel at different tasks, how do you decide when to use LogQuant or PartialLogQuant?

---

> ### Author Response · Authors · 2024-11-25
> **Response to Reviewer JWgv**
>
> ## For W3
> > The accuracy comparison only includes the quantization baseline KiVi but lacks an eviction-based baseline.
>
> In our initial study, we observed that eviction methods generally perform worse than quantization methods. To provide additional context, we conducted experiments comparing the attention error of LogQuant, KiVi, and LogQuant (Eviction) on Llama3.1-8B using the GSM8K dataset. For LogQuant the reserved length was set to \(42 \times 3 = 126\), while for KiVi, it was set to 128. The results are summarized in the table below:
>
> | Method                           | LogQuant | KiVi   | LogQuant (Eviction) | KiVi (Eviction) |
> | -------------------------------- | -------- | ------ | ------------------- | --------------- |
> | L1 error with original attention | **432.50**   | 556.10 | 1076.70             | 1612.56         |
>
> These results highlight the superior accuracy of quantization methods compared to eviction-based approaches.
>
> ---
>
> ## For Q1
> > Given that LogQuant and PartialLogQuant excel at different tasks, how do you decide when to use LogQuant or PartialLogQuant?
>
> This is an excellent question. **PartialLogQuant** was originally designed to address the observation that the Key Cache requires higher accuracy than the Value Cache.
>
> - In scenarios with a **sufficiently long reserved length**, the performance difference between **LogQuant** and **PartialLogQuant** is negligible.
> - However, **PartialLogQuant** is particularly advantageous in cases with **shorter reserved lengths** or **higher compression demands**, where it provides a more efficient trade-off between compression and accuracy.
>
> By tailoring the choice to the specific task requirements, we can maximize the overall performance of the model.

---

### Official Review · Reviewer_4LpZ · 2024-10-30

**Soundness:** 2
**Presentation:** 2
**Contribution:** 2
**Rating:** 3
**Confidence:** 3

**Summary:**

The paper presents LogQuant, a KV-cache quantization technique based on the observation that the positions of high-attention spikes follow a log distribution.

**Strengths:**

- The paper points out an interesting observation regarding the distribution of the positions of high-attention spikes.
- The paper targets the KV-cache quantization of LLM, which has a high importance in deploying LLM.

**Weaknesses:**

- The paper is mostly based on observation and proposes a heuristical solution.
- The presentation of LogQuant (mostly section 3) requires revision to clarify the solution better.

**Questions:**

- The authors use the terms “quantization” and “compression” alternatively, however, these terms have different meanings.
- I couldn’t find which quantization/compression technique is used in LogQuant. Furthermore, I found Figure 5 (and its caption) unclear.

---

> ### Author Response · Authors · 2024-11-22
> **Response to Reviewer 4LpZ**
>
> ## For Q2
> > I couldn’t find which quantization/compression technique is used in LogQuant. Furthermore, I found Figure 5 (and its caption) unclear.
>
> The quantization technique used in LogQuant is described in Section 3.4: *Implementation*. We employ **Quanto** as the quantization method. Additionally, Figure 5 provides an example to illustrate our method. We apologize if Figure 5 or its caption was unclear. We would greatly appreciate more specific feedback on which aspects of Figure 5 were difficult to interpret so we can improve its presentation in the revised version.

---

### Official Review · Reviewer_pDYU · 2024-11-03

**Soundness:** 2
**Presentation:** 3
**Contribution:** 3
**Rating:** 5
**Confidence:** 3

**Summary:**

This paper proposes a new KV-cache quantization algorithm called LogQuant.

The design of LogQuant relies on the empirical observation that the position of high-attention spikes follows a log distribution—namely, it becomes sparser as tokens move further from the current position. Accordingly, LogQuant keeps non-quantized cache entries with respect to the observed distribution and quantizes the rest to INT2.

The paper also proposes an interesting throughput optimization by observing that $Softmax(Q · K^T)$ and $V$ positions can be reordered without changing the computation's outcome.

Evaluation results over several LLMs and benchmarks show that LogQuant is more accurate than KiVi.

**Strengths:**

- The paper makes the interesting observation that high-attention positions follow a log-like distribution. This observation can help guide the design of future approaches.
-	The reordering observation is interesting and can help improve future approaches' inference speed.
-	The empirical results seem to set the new SOTA for 2-bit KV-cache quantization

**Weaknesses:**

-	Some design choices are not compelling:

    o	The quantization scheme of using INT2 is arbitrary and should be elaborated on.

    o	The design choice of which token to keep accurate is somewhat arbitrary. One can achieve a sparse pattern according to the desired distribution in different ways.

-	The evaluation leaves more to be desired:

    o	The prompt and generation lengths appear to be very small. Testing with larger context windows and generated sequences (e.g., 128K) would improve the paper.

    o	There is no evaluation of the quantization error of the tokens that are not quantized.

-	The degradation compared to the baseline is significant, and it is unclear where such a compromise can be acceptable.

**Questions:**

- What would the performance of LogQuant be if the quantized tokens were discarded completely?
- What group size is used in INT2? Is this overhead taken into account when comparing to other schemes?

- What about picking the retained accurate tokens in a way that preserves the distribution but non-deterministically?

- Will the accuracy of a model with LogQuant be better than that of a smaller baseline model?

---

> ### Author Response · Authors · 2024-11-22
> **Response to Reviewer pDYU**
>
> ## For W2
> > The prompt and generation lengths appear to be very small.
>
> We choose LongBench as one of our test dataset since it has a **maximum** input length of **32K**, which is the max position embedding limits for the model we chosed. We will add model with 128K length support in the future to improve our paper.
>
> ## For W3
> > The degradation compared to the baseline is significant, and it is unclear where such a compromise can be acceptable.
>
> We added a 4-bit result on **common response**, as shown that 4-bit quantization only has slight loss than baseline and can be acceptable, so what we want is to improve the accuracy on 2-bit mode.
>
> ## For Q1
> > What would the performance of LogQuant be if the quantized tokens were discarded completely?
>
> This is an insightful question. In our initial studies, we explored several eviction methods, such as streamingLLM and log sparse attention. These methods discard tokens directly, causing a change in the input sequence length and leading to information loss. As a result, they performed worse than quantization-based methods.
>
> To provide context, we conducted experiments comparing the attention error of LogQuant, KiVi, and LogQuant (Eviction) on Llama3.1-8B at GSM8K. For LogQuant, we set the reserved length as 42×3=126, and for KiVi, it was set to 128. The results are shown below:
>
> | Method                           | LogQuant | KiVi   | LogQuant (Eviction) | KiVi (Eviction) |
> | -------------------------------- | -------- | ------ | ------------------- | --------------- |
> | L1 error with original attention | **432.50**   | 556.10 | 1076.70             | 1612.56         |
>
> We also visualized this issue in a **Appendix A** of revised version, which demonstrates the problem more clearly.
>
> ## For Q2
> > What group size is used in INT2? Is this overhead taken into account when comparing to other schemes?
>
> We used a group size of 64 for all quantization methods. Details can be found in Section 4.1. Additionally, the associated overhead is accounted for in comparisons with other schemes.
>
> ## For Q3
> > What about picking the retained accurate tokens in a way that preserves the distribution but non-deterministically?
>
> Quantization is an irreversible compression method. Once a token is quantized, it cannot be restored to its original state unless recomputation is performed or the original data is stored elsewhere. As a result, it does not inherently reduce memory usage or improve performance.
>
> ## For Q4
> > Will the accuracy of a model with LogQuant be better than that of a smaller baseline model?
>
> Yes, it is possible for a model using LogQuant to outperform a smaller baseline model in accuracy. We believe the primary determinant of LLM performance is the model size. For instance, as shown in Table 2's Multi-Document QA tasks, Qwen1.5-7B with a baseline cache achieves an accuracy of 33.05, while Qwen1.5-14B with LogQuant achieves a higher accuracy of 38.01.

---

> > ### Comment · Reviewer_pDYU · 2024-11-26
> >
> > I acknowledge the authors' rebuttal and will keep my score.

---

### Official Review · Reviewer_cG5f · 2024-11-03

**Soundness:** 2
**Presentation:** 2
**Contribution:** 2
**Rating:** 3
**Confidence:** 4

**Summary:**

The paper presents LogQuant, a new 2-bit quantization approach for compressing the KV cache in LLM inference. This approach applies a log-based filtering mechanism that enables significant memory savings while preserving performance. Unlike traditional methods that prioritize recent tokens or rely on predicted attention patterns, LogQuant uses a log-distributed approach to selectively compress tokens. LogQuant shows improvements in throughput (by 25%) and batch size (by 60%), and reportedly improves task accuracy for complex tasks like math and code completion by 40-200% at similar compression ratios.

**Strengths:**

- This work is based on interesting findings of attention score distribution among token positions.
- Leveraging a log distribution for token selection is innovative and addresses a core limitation in existing KV cache compression methods, improving the balance between memory use and performance.
- The method’s compatibility with popular inference frameworks makes it easily adaptable.

**Weaknesses:**

- The evaluation only compares with KIVI [1] on task performance, which lacks a broad comparison with other compression methods. A more comprehensive range of baseline methods—like KVQuant [2], SKVQ [3], etc. Other types of compression methods [4, 5] under similar compression rates can be included. Also compression settings like 4-bit quantization—would provide a fuller view of its strengths and trade-offs.
- For many tasks, LogQuant results in a substantial accuracy drop (10+ points in some cases), which raises concerns about its reliability in sensitive tasks.
- This work does not sufficiently discuss the overhead from operations like slicing and concatenating.
- In many cases, the model experiences unexpectedly large accuracy drops, compared to numbers in other KV cache compression methods [1,2,3].

[1] Kivi: A tuning-free asymmetric 2bit quantization for kv cache

[2] Kvquant: Towards 10 million context length llm inference with kv cache quantization

[3] SKVQ: Sliding-window Key and Value Cache Quantization for Large Language Models

[4] Flexgen: High-throughput generative inference of large language models with a single gpu

[5] Atom: Low-bit quantization for efficient and accurate llm serving

**Questions:**

- Could you expand the comparisons to include other compression strategies, especially those operating at similar compression rates? And what are the performance results for LogQuant under 4-bit quantization?
- Could you provide discussion and profiling of the overhead from additional operations, such as slicing and concatenating, to quantify their impact on throughput?
- What are the results on models like LLaMA3-8B (3.0), Mistral, and Lonchat? Refer to settings of https://github.com/henryzhongsc/longctx_bench [1].

[1] Kv cache compression, but what must we give in return? a comprehensive benchmark of long context capable approaches

---

> ### Author Response · Authors · 2024-11-25
> **Response to Reviewer cG5f**
>
> ## For Q1 and W2
> > For many tasks, LogQuant results in a substantial accuracy drop (10+ points in some cases), which raises concerns about its reliability in sensitive tasks.
> > Could you expand the comparisons to include other compression strategies, especially those operating at similar compression rates? And what are the performance results for LogQuant under 4-bit quantization?
>
> We have added the results for **4-bit quantization** in the **Common Responses** section. In 4-bit quantization, both KiVi and LogQuant achieve performance very close to the baseline. However, our focus is on improving the performance under 2-bit quantization, where the loss is more significant.
>
> As shown in the table, while LogQuant may result in accuracy drops in some sensitive tasks, it still consistently outperforms KiVi, even under challenging compression scenarios.
>
> ---
>
> ## For Q3
> > What are the results on models like LLaMA3-8B (3.0), Mistral, and Lonchat? Refer to settings of https://github.com/henryzhongsc/longctx_bench [1].
>
> For **LLaMA3-8B (3.0)**, please refer to the first row of **Table 4** in **Appendix A**. Currently, we have tested LLaMA, Qwen, and Phi-3 using their official chat templates.
>
> Our evaluation settings align with the official **LongBench** evaluation code. Additionally, the model output **metadata** and the testing pipeline are provided in the **Supplementary Material** for further reference.

---

### Official Review · Reviewer_H2zG · 2024-11-04

**Soundness:** 2
**Presentation:** 2
**Contribution:** 1
**Rating:** 3
**Confidence:** 4

**Summary:**

This paper proposes LogQuant, a KV cache quantization method for improving the memory efficiency and throughput of LLM inference.  Previous methods, such as KIVI and StreamingLLM, assume the most recent tokens and the first few sink tokens are more important for model performance. This work observes using log-distributed recent tokens are better for preserving model accuracy. Hence, the authors propose LogQuant, which stores a set of tokens at log-distributed positions in full precision, while keeping all other tokens quantized to achieve KV cache compression. Empirical evaluations show that LogQuant outperforms KIVI at 2-bit quantization.

**Strengths:**

1. This paper studies an important problem.
2. The presentation, including the figures and tables, are overall good.

**Weaknesses:**

1. The proposed method lacks novelty. The proposed LogQuant is highly similar to KIVI: they both use integer quantization with a fixed-sized full-precision cache, and the only difference is the selection method for the full-precision tokens. This work is also similar to mixed-precision approach for KV cache quantization such as [1,2], which identify outlier tokens in the KV cache and preserve in higher precision or full precision.
2. The token selection process of the full-precision cache in LogQuant is fixed and non-adaptive. The token selection is determined only by token position, and not dependent on attention score or token importance. As the authors illustrate in Figure 2, the outliers in attention score do not follow a fixed pattern. Hence, it is questionable whether using a fixed pattern of full-precision cache improves the accuracy of  KV cache quantization universally for all downstream tasks.
3. The experiments are not comprehensive. 4-bit quantization offers better quality than 2-bit quantization, and it is missing from the experiments. The baseline KIVI is also missing in the memory usage and throughput comparison in Figure 7.

References

[1] He, Yefei, et al. "ZipCache: Accurate and Efficient KV Cache Quantization with Salient Token Identification." NeurIPS 2024.

[2] Dong, Shichen, et al. "QAQ: Quality Adaptive Quantization for LLM KV Cache." arXiv preprint arXiv:2403.04643 (2024).

**Questions:**

1. How does LogQuant compare with KIVI for 4-bit quantization? And how do they compare in terms of inference latency and memory usage?
2. In Table 2, is LogQuant achieving better accuracy than KIVI using equal or less memory budget?

---

> ### Author Response · Authors · 2024-11-22
> **Response to Reviewer H2zG**
>
> ## For W3 and Q1
> In 4-bit quantization, most methods achieve performance very close to the baseline, so our focus shifts to the more challenging 2-bit quantization. In this setting, LogQuant demonstrates significantly better performance compared to KiVi. For reference, we have also included a comparison with the 4-bit mode in the **Common Responses**.
>
> ## For W2
> > The token selection process of the full-precision cache in LogQuant is fixed and non-adaptive.
>
> We address this concern in Section 3.1, where we explain that the attention pattern is inherently dynamic, while quantization is irreversible. Approaches such as QAQ [2] necessitate offloading the original precision KV cache to the CPU to dynamically adjust token precision stored in GPU memory. In contrast, without this offloading, the quantized positions must be fixed, as additional storage to retain adaptability is not feasible.
>
> ##  For Q2
> > In Table 2, is LogQuant achieving better accuracy than KIVI using equal or less memory budget?
>
> As discussed of reserved length configuration in Section 4.1, all results achieved by LogQuant use **equal** or less memory budget compared to KIVI, as detailed in the configuration.

---

> > ### Comment · Reviewer_H2zG · 2024-12-02
> >
> > Thank you to the authors for their response. However, my concern regarding the fixed and non-adaptive token selection approach in LogQuant remains unaddressed. While the paper demonstrates empirical improvements over KIVI in the 2-bit quantization region, there is no significant difference observed in the 4-bit region. Furthermore, the contributions presented in the paper appear incremental, primarily building upon previous works rather than introducing substantial novelty. Therefore, I have decided to maintain my original score.

---

### Author Response · Authors · 2024-11-22
**Common Responses**

We would like to thank all reviewers for their insightful comments and constructive feedback, which have greatly contributed to the improvement of our work. Below, we address some common questions raised by the reviewers.

## Common Question on 4-bit Results
Several reviewers raised concerns about the lack of testing with 4-bit quantization. Initially, we did test the 4-bit quantization and found that the loss in accuracy was minimal. As a result, we decided to focus on 2-bit quantization for the initial study. However, we have now included both 2-bit and 4-bit results for both KiVi and LogQuant, as well as the baseline on LongBench with Llama3.1-8B, to provide a more comprehensive comparison. The updated meta-data will be revised in the Appendix for better clarity.

| **Category**          | **KiVi (2-bit)** | **KiVi (4-bit)** | **LogQuant (2-bit)** | **LogQuant (4-bit)** | **Baseline** |
|-----------------------|------------------|------------------|----------------------|----------------------|--------------|
| Single-Document QA     | 38.89            | 47.75            | 41.91                | 47.73                | 47.71        |
| Multi-Document QA      | 34.02            | 39.74            | 36.08                | 39.93                | 39.96        |
| Summarization         | 16.10            | 17.94            | 16.62                | 17.92                | 18.08        |
| Few-shot Learning     | 52.51            | 61.34            | 56.43                | 61.21                | 61.22        |
| Synthetic Tasks       | 45.02            | 67.74            | 52.51                | 67.68                | 67.78        |
| Code Completion       | 43.06            | 59.53            | 52.10                | 59.57                | 59.78        |

---

### Meta-Review · Area_Chair_yWLD · 2024-12-21

**Metareview:**

This paper proposes a 2-bit quantization method for KV cache compression in large language model inference, aiming to improve memory efficiency and computational performance. While the method demonstrates interesting insights, such as leveraging a log-distributed filtering mechanism for token compression, its novelty is limited compared to prior work. The token selection process is fixed and non-adaptive, which raises concerns about its generalizability across various tasks. Additionally, the evaluation lacks comprehensive comparisons with a broader set of baselines and critical metrics such as latency, real-world performance with longer context windows, and quantization error. Although the authors provided supplementary data for 4-bit quantization in their rebuttal, the results showed incremental improvements without addressing the core limitations.
The primary reasons for rejection include insufficient novelty, a lack of rigorous evaluation, and limited applicability of the proposed method. These issues suggest that the paper does not meet the standards for publication at current venue.

**Additional Comments On Reviewer Discussion:**

The reviewers collectively highlighted the paper's incremental contributions, limited baseline comparisons, and lack of evaluation on key metrics. While the authors responded to some concerns, the core issues regarding the method’s adaptiveness and comprehensiveness were not sufficiently addressed, leading to the decision to reject the submission.

---

### Decision · Program_Chairs · 2025-01-22

Reject